# Differentiating between *Enterococcus*
*faecium* and *Enterococcus*
*lactis* by Matrix-Assisted Laser Desorption Ionization Time-of-Flight Mass Spectrometry

**DOI:** 10.3390/foods11071046

**Published:** 2022-04-05

**Authors:** Eiseul Kim, Seung-Min Yang, Hyun-Jae Kim, Hae-Yeong Kim

**Affiliations:** Department of Food Science and Biotechnology, Institute of Life Sciences & Resources, Kyung Hee University, Yongin 17104, Korea; eskim89@khu.ac.kr (E.K.); ysm9284@gmail.com (S.-M.Y.); tkdzmtm2@naver.com (H.-J.K.)

**Keywords:** *Enterococcus faecium*, *Enterococcus lactis*, MALDI-TOF MS, identification, differentiation, specific mass peak, in-house database

## Abstract

Unlike *Enterococcus faecium* strains, some *Enterococcus lactis* strains are considered potential probiotic strains as they lack particular virulence and antibiotic resistance genes. However, these closely related species are difficult to distinguish via conventional taxonomic methods. Here, for the first time, we used matrix-assisted laser desorption/ionization time-of-flight mass spectrometry (MALDI-TOF MS) with BioTyper and in-house databases to distinguish between *E*. *faecium* and *E*. *lactis*. A total of 58 reference and isolated strains (89.2%) were correctly identified at the species level using MALDI-TOF MS with in-house databases. However, seven strains (10.8%) were not accurately differentiated as a single colony was identified as a different species with a similar score value. Specific mass peaks were identified by analyzing reference strains, and mass peaks at 10,122 ± 2 *m*/*z*, 3650 ± 1 *m*/*z*, and 7306 ± 1 *m*/*z* were unique to *E*. *faecium* and *E*. *lactis* reference strains, respectively. Mass peaks verified reproducibility in 60 isolates and showed 100% specificity, whereas 16S rRNA sequencing identified two different candidates for some isolates (*E*. *faecium* and *E*. *lactis*). Our specific mass peak method helped to differentiate two species, with high accuracy and high throughput, and provided a viable alternative to 16S rRNA sequencing.

## 1. Introduction

Enterococci belong to the lactic acid bacterium and are usually present in plant materials and vegetables, especially raw milk or dairy products [1]. Previous microbiota studies in fermented foods reported that enterococci have important roles in fermentation and contribute to the unique taste and flavor of fermented foods [1]. Moreover, enterococci can also improve hygiene and safety in some foods as they produce antimicrobial substances such as bacteriocins (enterocins) or lactic acid [1]. Enterococci, especially *Enterococcus faecalis* and *Enterococcus faecium*, have great potential as probiotics, yet, some strains are associated with human infection, virulence factors, and antibiotic resistance, including resistance to vancomycin [1,2]. However, *E*. *lactis*, which is closely related to *E*. *faecium*, lacks hospital infection-associated markers, such as insertion sequence *IS16* and glycosyl hydrolase *hyl_Efm_*, suggesting *E*. *lactis* complies with European Food Safety Authority guidelines [3]. Therefore, *E*. *lactis* displays a higher potential as a probiotic strain than *E*. *faecium*, as the absence of transferable virulence and antibiotic resistance genes is an important prerequisite when screening probiotic strains [4].

Scientists have conventionally relied on physiological and biochemical properties to identify lactic acid bacteria [5]. However, *Enterococcus* species share many characteristics, making conventional identification methods not only inaccurate but also time-consuming. Recently, whole-genome sequencing was applied to bacterial taxonomy and successfully discriminated closely related species, including *E*. *faecium* and *E*. *lactis* [3]. However, this technique is time-consuming, expensive, and requires additional analysis steps, such as average nucleotide identity or digital DNA-DNA hybridization. Therefore, routine use in the laboratory is difficult [6]. Currently, 16S rRNA sequencing is a commonly used molecular method to classify bacteria. Strains showing more than 98.7% sequence similarity in 16S rRNA genes are considered the same species [7]. Unfortunately, poor discrimination has been reported for *Enterococcus* due to high sequence similarities (99%) in 16S rRNA [3]. By contrast, protein-coding genes provide higher taxonomic resolution and could serve as alternatives to 16S rRNA sequencing in discriminating closely related species [6,8].

Matrix-assisted laser desorption/ionization time-of-flight mass spectrometry (MALDI-TOF MS) is often used to identify and differentiate microorganisms [9,10]. The approach is rapidly replacing analytical phenotypic and conventional biochemical identification methods, especially in clinical microbiology laboratories [11,12]. This method has been successfully used in clinical diagnostic settings and has been expanded into food safety, fermented food monitoring, biodiversity, and gut microbiota research [13,14,15,16]. Generally, MALDI-TOF MS distinguishes at the species level, with taxonomic resolution observed at the subspecies or serovar level when combined with specific mass peaks [17,18]. Importantly, MALDI-TOF MS accuracy depends on a reference microorganism database. However, commercial databases are mainly designed for routine clinical diagnostics; therefore, adding additional entries to such databases are important to facilitate increased identification rates.

In this study, we used MALDI-TOF MS to identify and discriminate between *E*. *faecium* and *E*. *lactis*. The BioTyper database currently lacks *E*. *lactis* reference spectra. Thus, we constructed an in-house database coupled with specific mass peaks to compare data with 16S rRNA sequencing.

## 2. Materials and Methods

### 2.1. Enterococcus Isolates and Growth Conditions

To generate main spectrum profiles, 25 *Enterococcus* reference strains, of which *E*. *avium* (n = 2), *E*. *casseliflavus* (n = 1), *E*. *devriesei* (n = 1), *E*. *durans* (n = 1), *E*. *faecalis* (n = 3), *E*. *faecium* (n = 2), *E*. *gallinarum* (n = 1), *E*. *gilvus* (n = 1), *E*. *hirae* (n = 4), *E*. *lactis* (n = 3), *E*. *malodoratus* (n = 1), *E*. *mundtii* (n = 2), *E*. *pseudoavium* (n = 1), *E*. *raffinosus* (n = 1), and *E*. *saccharolyticus* (n = 1), were used (Table 1). All strains were obtained from the Korean Collection for Type Cultures (KCTC, Daejeon, Korea), the Korean Agricultural Culture Collection (KACC, Jeonju, Korea), and the National Culture Collection for Pathogens (NCCP, Ceongju, Korea). Reference strains were grown under anaerobic conditions on de Man, Rogosa, and Sharpe agar (MRS, Difco, Becton & Dickinson, Sparks, MD, USA) for 48 h at 37 °C [19,20,21].

To verify main spectrum profiles, bacteria from fermented foods, such as soybean paste, soy sauce, sikhae, and raw milk were isolated according to a previous study [22]. Briefly, 25 g of each food sample was homogenized in 225 mL sterile phosphate-buffered saline and serially diluted. Then, 0.1 mL of each dilution was spread onto MRS agar plates and incubated at 37 °C for 48 h. Isolates were identified using 16S rRNA sequencing via the 27F/1492R primer set. Isolates other than *E*. *faecium* and *E*. *lactis* were excluded from the research.

### 2.2. 16S rRNA Sequencing

The 16S rRNA sequencing of isolates was performed to compare the MALDI-TOF MS results. Genomic DNA of isolates was extracted using G-spin genomic DNA extraction kit (Intron Biotechnology, Seongnam, Korea). The amplification was carried out in a 25 µL mixture containing 2.5 mM dNTPs (Takara, Tokyo, Japan), 10× buffer (Takara, Tokyo, Japan), 0.5 units *Ex Taq* polymerase (Takara, Tokyo, Japan), 20 ng of template, and 400 nM of 27F/1492R primer set. The PCR thermal profile was performed at 95 °C for 5 min, followed by 30 cycles of 95 °C for 1 min, 58 °C for 1 min, and 72 °C for 2 min, and concluded with a final elongation at 72 °C for 10 min. The PCR product was purified using the QIAquick PCR purification kit (Qiagen, Hilden, Germany) and sequenced. The 16S rRNA sequences of isolates were analyzed using the BLAST program (NCBI, Bethesda, MD, USA).

### 2.3. Identifying Specific Mass Peaks

#### 2.3.1. Sample Preparation for MALDI-TOF MS

Protein from reference strains was extracted using an existing ethanol/formic acid protocol [23]. Briefly, 10 µL fresh culture was suspended in 300 µL water and mixed with 900 µL ethanol to inactivate the bacteria. The cell suspension was then centrifuged at 13,600× *g* for 10 min and supernatant was removed. Once dry, the pellet was resuspended in 20 µL 70% formic acid and 20 µL acetonitrile, and centrifuged at 13,600× *g* for 5 min. After this, 1 µL extract was spotted onto an MSP 96 polished steel target plate (Bruker Daltonics, Bremen, Germany) and air dried for 10 min. Spots were overlaid with 1 µL α-cyano-4-hydroxycinnamic acid (CHCA) matrix solution (Bruker Daltonics, Bremen, Germany), and air dried for sample/matrix cocrystallization.

#### 2.3.2. MALDI-TOF MS Analysis

Analyses were performed via a Microflex LT bench-top mass spectrometer (Bruker Daltonics, Bremen, Germany) with FlexControl software version 3.4. Data were obtained in automatic mode by collecting 240 laser shots with 40% laser intensity. Spectra were recorded in a positive linear mode (ion source one voltage = 18.00 kV; ion source two voltage = 16.38 kV; lens voltage = 5.40 kV; laser frequency = 60 Hz; and mass range = 2000–20,000 Da). Calibration and quality control steps before strain identification were performed via a bacterial test standard (Bruker Daltonics, Bremen, Germany) which consisted of an *Escherichia coli* DH5-α protein extract.

To identify isolates with specific mass peaks, raw spectra were normalized, and strain peak areas and intensities were analyzed via FlexAnalysis software version 3.4 (Bruker Daltonics, Bremen, Germany). Then, isolates were identified by comparing the presence or absence of species-specific mass peaks. A main spectrum profile dendrogram and principal component analyses (PCA) for reference and isolate strains were performed via MALDI BioTyper software version 3.1 (Bruker Daltonics, Bremen, Germany) as per standard operating procedures.

### 2.4. Creating an In-House Database

A representative *E*. *lactis* reference strain was used to construct an in-house database. Main spectra were generated as described in Section 2.2 identifying specific mass peaks. In total, 30 replicates for the *E*. *lactis* strain were incorporated. Raw spectra quality was evaluated using FlexAnalysis software version 3.4 (Bruker Daltonics, Bremen, Germany), whereby spectra displaying high background noise were deleted [24] according to the manufacturer’s instructions. After baseline subtraction and smoothing, >20 high-quality spectra were selected and transferred to create the main spectrum profile which was used for in-house database supplementation. The 25 reference strains were blindly evaluated to determine mass spectra reproducibility with MALDI-TOF MS identification. The in-house database was assessed based on 60 isolate measurements.

### 2.5. Identifying Isolates Using Specific Peaks

To identify isolates, proteins were extracted using the extended direct transfer extraction protocol [25]. Briefly, a single bacterial colony was spotted on the MSP 96 polished steel target plate and overlaid with 1 µL 70% formic acid. After drying, the area was covered with 1 µL CHCA matrix solution (Bruker Daltonics, Bremen, Germany). The plate was loaded into the Microflex LT bench-top mass spectrometer which contained the BioTyper database version 3.4 (5627 reference spectra) and the in-house database, and then analyzed as described. The MALDI-TOF MS analysis results are generally expressed with a score value, indicative of the matching between the sample spectrum and the reference spectra in database. Score results were between 0 to 3. The identification criteria were as follows: a score of ≥2.300 was considered as high probable species level; 2.000–2.299, a probable species identification; 1.700–1.999, a probable genus identification; and <1.700, no reliable identification.

## 3. Results and Discussion

### 3.1. Identifying Specific Mass Peaks

It was previously reported that reliance on commercial databases could yield ambiguous results for closely related bacterial species, such as *Lactobacillus johnsonii* and *Lactobacillus gasseri*, *Lactiplantibacillus plantarum*, and *Lactiplantibacillus paraplantarum*, and *Bacillus punilus* and *Bacillus safensis* [25,26,27]. Importantly, MALDI-TOF MS combined with specific mass peaks was successfully used to discriminate between closely related species or subspecies, including *Lactobacillus paracasei* subspecies, *Bifidobacterium animalis* subspecies, *Streptococcus* species, and *Lactiplantibacillus* species [6,18,25,28,29]. Therefore, the characterization of specific mass peaks for species identification is accepted. In the present study, we observed inaccurate or ambiguous identification between *E*. *faecium* and *E*. *lactis* in the MALDI database.

Mass spectra showed similar patterns between *E*. *faecium* and *E*. *lactis* (Figure 1). The mass spectra of each analyzed strain for non-target species are shown in Appendix A. Discrimination ability at the species level was evaluated by analyzing mass peaks from five reference strains and 20 reference strains comprising 13 different species. In total, 192 mass peaks were extracted from the mass spectra of five reference *E*. *faecium* and *E*. *lactis* strains and analyzed for peak values according to species. Moreover, specific mass peaks were compared with 943 mass peaks from 13 other species to confirm they were unique peaks and not found in other species.

In *E*. *faecium*, a mass peak at 10,122 ± 2 *m*/*z* was common to all *E*. *faecium* strains; peaks were present in two *E*. *faecium* reference strains but absent in other *Enterococcus* species, including *E*. *lactis* (Table 2). In total, 15 mass peaks were common in *E*. *lactis* strains; *E*. *lactis* was characterized by mass peaks at 3650 ± 1 *m*/*z* and 7306 ± 1 *m*/*z* which were not identified in the other 14 species, including *E*. *faecium* (Table 2). Therefore, mass peaks at 10,122 ± 2 *m*/*z* were unique to *E*. *faecium*, 3650 ± 1 *m*/*z* and 7306 ± 1 *m*/*z* were uniquely found in *E*. *lactis* (Figure 2).

### 3.2. Evaluating Commercial and In-House Databases

We used the BioTyper database to evaluate species differentiation between *E*. *faecium* and *E*. *lactis*. Five reference strains and 60 isolates were tested via BioTyper and in-house databases. Reference strains included two *E*. *faecium* (KACC 11954 and KCTC 13225) and three *E*. *lactis* strains (KACC 15681, KACC 14552, and KACC 21015). As a result, 65 colonies were identified as *E*. *faecium* via the BioTyper database. Of these, six strains (9.2%) were identified at the highly probable species level (score ≥ 2.300), 53 strains (81.5%) were identified at the probable species level (2.000–2.299), and the remaining six (9.2%) were identified at the probable genus level (1.700–1.999) (Table 3). All isolates were identified as *E*. *faecium* at the species level via the BioTyper database.

*E*. *lactis* did not exist in the BioTyper database and was created in the in-house database. After generating *E*. *lactis* strain spectra, five reference strains and 60 isolates were re-identified. The 59 strains (90.8%) were correctly identified with a high score value ≥ 2.300, and six strains (9.2%) were identified at the probable species level (2.000–2.299) (Table 3). The in-house database, with added *E*. *lactis* mass spectra, accurately identified 58 strains (89.2%), generating an improved identification rate when compared with the BioTyper database, but seven strains (10.8%) had unreliable results due to spectral similarity with *E*. *faecium*. All strains were identified as *E*. *faecium* (5/12, 41.7%) and *E*. *lactis* (53/53, 100%) at the species level, whereas some *E*. *faecium* strains (7/12, 58.3%) generated unreliable results (Table 3 and Table 4). Seven isolates were identified as *E*. *lactis* in the first match, with score values between 2.205 and 2.413, but the second match identified *E*. *faecium*, with score values between 2.160 and 2.370. Therefore, these isolates could not be differentiated by both BioTyper and in-house databases.

BioTyper database limitations were also previously observed for *Lactiplantibacillus* species, *Salmonella* species, and some anaerobic bacteria [23,25,30,31]. These species are phylogenetically closely related and have similar protein mass spectra; therefore, they could not be accurately distinguished by this database. A previous study also reported that an improved commercial database facilitated the accurate identification of microorganisms from a single colony [18]. However, in our study, the expanded database improved the identification rate between two species, but could not clearly distinguish all strains due to high similarity between protein mass spectra. To differentiate these spectra, re-identification is required using additional tests based on isolated characteristics [24].

Five reference strains and 60 isolates were used to evaluate MALDI-TOF MS robustness. The main spectrum profile dendrogram and PCA clustering are practical for differentiating between closely related strains and determining associations between isolated strains [32]. Dendrogram and PCA clustering were performed to confirm the discriminative power of mass spectra to identify two species; all *E*. *faecium* and *E*. *lactis* strains were classified into two distinct groups in the dendrogram (Figure 3). The first cluster included *E*. *faecium* species, and two clusters included *E*. *lactis*. PCA clustering was performed using intensities and mass values and showed both species were separated (Figure 4). This result suggests that two species may be differentiated by mass spectra obtained with MALDI-TOF MS.

### 3.3. Identifying Isolates Using Specific Mass Peaks

To validate our *E*. *faecium* and *E*. *lactis* identification approach, 60 isolates were identified using specific mass peaks; these peaks in type strains were consistently identified in isolates. Ten isolates were identified as *E*. *faecium* via mass peak analysis (Table 4). The peak at 10,122 ± 2 *m*/*z* was specific to *E*. *faecium* and was uniquely present in these isolates, whereas *E*. *lactis* mass peaks were absent. All isolates were consistent with 16S rRNA sequencing identification results (Table 5). These isolates were identified as *E*. *faecium* (accession no. FJ378693.1 or MN401132.1 or MH473158.1) via 16S rRNA sequencing.

The 50 isolates were identified as *E*. *lactis*; the mass peaks at 10,122 ± 2 *m*/*z* and 3650 ± 1 *m*/*z*, specific to *E*. *lactis*, were present in these isolates, but specific *E*. *faecium* peaks were absent. These isolates were then compared with 16S rRNA sequencing results. Isolates were correctly identified as one species using mass peak analysis, whereas 16S rRNA sequencing generated two different species candidates, *E*. *faecium* (accession no. MT597585.1 or MT378127.1) and *E*. *lactis* (MG948154.1 or CP082267.1), instead of one species. These data were consistent with previous studies showing that 16S rRNA sequence analyses showed limited discriminatory power between *E*. *faecium* and *E*. *lactis*, as both exhibited >99% sequence homology [1,3]. Therefore, three mass peaks were specific for identifying and discriminating between *E*. *faecium* and *E*. *lactis*.

MALDI-TOF MS is a cost-efficient and rapid identification method when compared to other techniques [33]. The approach was used to identify ten strains within 15 min in a colony selection study [6]. The higher the throughput rate of a sample is, the lower the analysis cost/isolate [34]. To efficiently identify microorganisms, MALDI-TOF MS costs do not exceed $0.2 per strain, whereas other identification approaches, such as polymerase chain reaction-based methods, are more expensive [6,33].

Our identification method rapidly and accurately detected *E*. *faecium* and *E*. *lactis* from MALDI-TOF MS-specific mass peaks. *E*. *faecium* and *E*. *lactis* strains were not correctly identified at the species level using an in-house database; however, they were confirmed and identified by mass peak analysis. Despite the fact the in-house database misidentified a high number of isolates (10.8%), peak analyses may facilitate correct species assignment. The approach may also save on sequencing costs, and it does not require sequence amplification and genomic DNA extraction, thereby reducing costs, time, and labor for final strain identification [6]. Also, the extended direct transfer extraction protocol was used to reduce protein extraction times and shorten turnaround times. The specific mass peaks identified in this study were successfully used to identify *E*. *faecium* and *E*. *lactis* strains; therefore, this approach could be considered more efficient and accurate than 16S rRNA sequencing which is lacking in discriminating power.

## 4. Conclusions

MALDI-TOF MS is a powerful tool that distinguishes between *E*. *faecium* and *E*. *lactis* species. Moreover, the identification based on mass spectrometric data of two species, by combining an in-house database and MALDI-TOF MS-specific mass peak data, showed a better discrimination power than 16S rRNA sequencing. This approach can be successfully used for the accurate, rapid identification, and discrimination of *E*. *faecium* and *E*. *lactis* species and could be used in quality control protocols in the probiotic industry.

## Figures and Tables

**Figure 1 foods-11-01046-f001:**
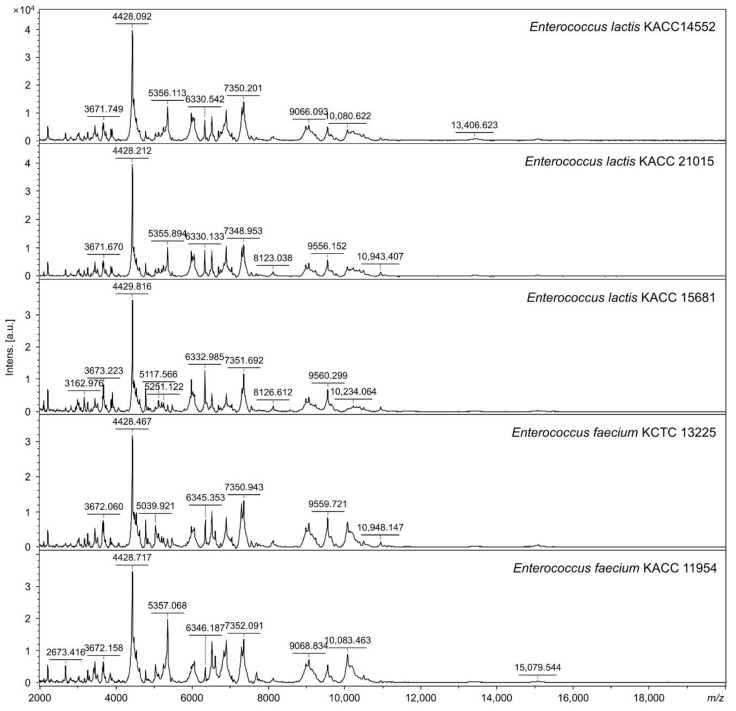
Mass spectra of reference strains of *E*. *lactis* KACC 14552, *E*. *lactis* KACC 15681, *E*. *lactis* KACC 21015, *E*. *faecium* KACC 11954, and *E*. *faecium* KCTC 13225; *m/z*, mass-to-charge ratio; a.u., arbitrary units.

**Figure 2 foods-11-01046-f002:**
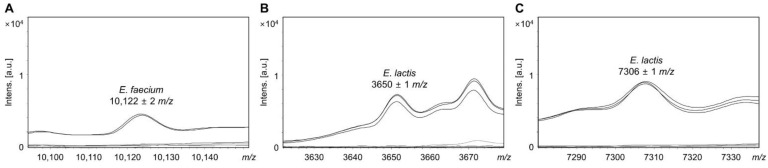
Specific mass peaks for *E*. *faecium* and *E*. *lactis*. (**A**) Mass peak at 10,122 ± 2 *m*/*z* present in *E*. *faecium* strains, (**B**) mass peak at 3650 ± 1 *m*/*z* present in *E*. *lactis* strains, and (**C**) mass peak at 7306 ± 1 *m*/*z* present in *E*. *lactis* strains. Figure was generated using FlexAnalysis software version 3.4 (Bruker Daltonics, Bremen, Germany).

**Figure 3 foods-11-01046-f003:**
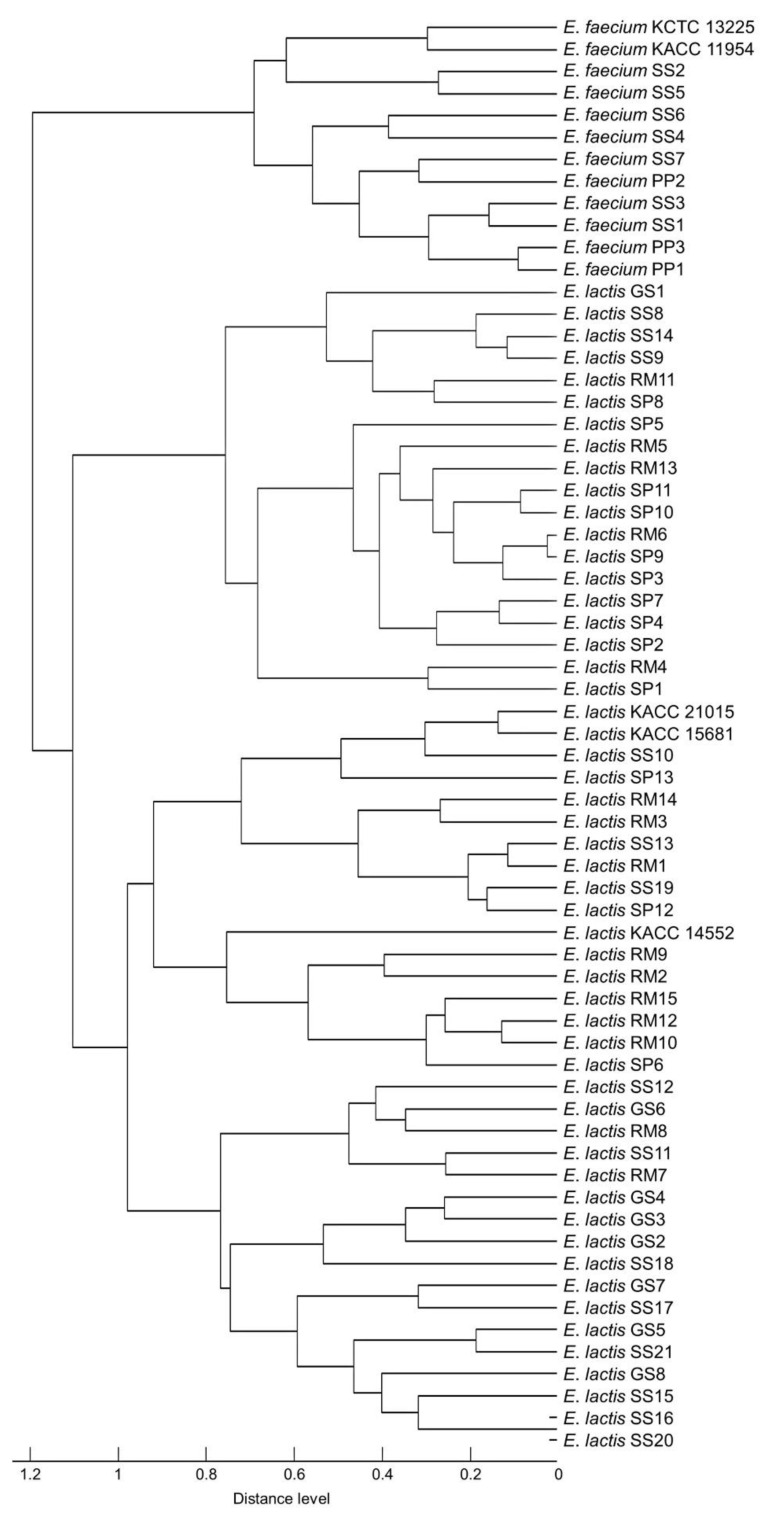
Main spectrum profiles dendrogram generated based on *m*/*z* value and relative intensities of five reference strains and 60 isolates.

**Figure 4 foods-11-01046-f004:**
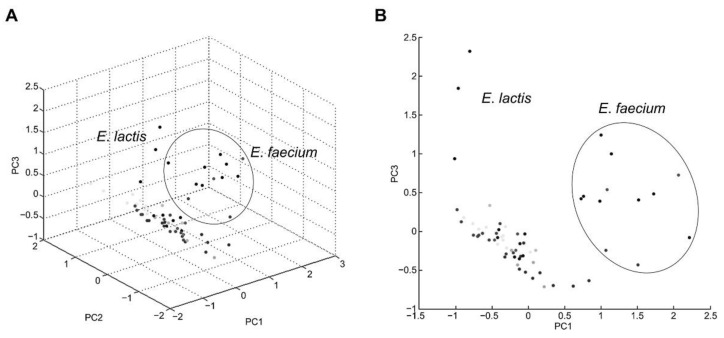
Principal component analysis (PCA) generated by mass spectra of five reference strains and 60 isolates. Each dot on the (**A**) three-dimensional plot and (**B**) two-dimensional plot represent strains. Dots included a in circle represent *E*. *faecium* strains.

**Table 1 foods-11-01046-t001:** List of reference strains used in this study.

Bacterial Strains	Origins
Reference strains	
*Enterococcus avium*	KACC 10788
*Enterococcus avium*	NCCP 10761
*Enterococcus casseliflavus*	KCTC 3552
*Enterococcus devriesei*	KACC 14590
*Enterococcus durans*	KCTC 13289
*Enterococcus faecalis*	KACC 11859
*Enterococcus faecalis*	KCTC 3206
*Enterococcus faecalis*	KCTC 5290
*Enterococcus faecium*	KACC 11954
*Enterococcus faecium*	KCTC 13225
*Enterococcus gallinarum*	NCCP 11518
*Enterococcus gilvus*	KACC 13847
*Enterococcus hirae*	KACC 10779
*Enterococcus hirae*	KACC 10782
*Enterococcus hirae*	KACC 13884
*Enterococcus hirae*	KACC 16328
*Enterococcus lactis*	KACC 14552
*Enterococcus lactis*	KACC 15681
*Enterococcus lactis*	KACC 21015
*Enterococcus malodoratus*	KACC 13883
*Enterococcus mundtii*	KACC 13824
*Enterococcus mundtii*	KCTC 3630
*Enterococcus pseudoavium*	KACC 13781
*Enterococcus raffinosus*	KACC 13782
*Enterococcus saccharolyticus*	KACC 10789
Isolates (no. of isolates)	
SP1-SP13 (13)	Soybean paste
RM1-RM15 (15)	Raw milk
PP1-PP3 (3)	Probiotic product
SS1-SS21 (21)	Soy sauce
GS1-GS8 (8)	Gajami-sikhae

KACC, Korean Agricultural Culture Collection; NCCP, National Culture Collection for Pathogens; KCTC, Korean Collection for Type Cultures.

**Table 2 foods-11-01046-t002:** Presence/absence of specific mass peaks for 25 reference strains.

Strains	MALDI-TOF MS Identification		Mass Peak (*m*/*z*)		
	BioTyper	In-House Database	10,122 ± 2	3650 ± 1	7306 ± 1
KACC 11954	*E*. *faecium*	*E*. *faecium*	+	−	−
KCTC 13225	*E*. *faecium*	*E*. *faecium*	+	−	−
KACC 14552	*E*. *faecium*	*E*. *lactis*	−	+	+
KACC 15681	*E*. *faecium*	*E*. *lactis*	−	+	+
KACC 21015	*E*. *faecium*	*E*. *lactis*	−	+	+
KACC 10788	*E*. *avium*	*E*. *avium*	−	−	−
NCCP 10761	*E*. *avium*	*E*. *avium*	−	−	−
KCTC 3552	*E*. *casseliflavus*	*E*. *casseliflavus*	−	−	−
KACC 14590	*E*. *devriesei*	*E*. *devriesei*	−	−	−
KCTC 13289	*E*. *durans*	*E*. *durans*	−	−	−
KACC 11859	*E*. *faecalis*	*E*. *faecalis*	−	−	−
KCTC 3206	*E*. *faecalis*	*E*. *faecalis*	−	−	−
KCTC 5290	*E*. *faecalis*	*E*. *faecalis*	−	−	−
NCCP 11518	*E*. *gallinarum*	*E*. *gallinarum*	−	−	−
KACC 13847	*E*. *gilvus*	*E*. *gilvus*	−	−	−
KACC 10779	*E*. *hirae*	*E*. *hirae*	−	−	−
KACC 10782	*E*. *hirae*	*E*. *hirae*	−	−	−
KACC 13884	*E*. *hirae*	*E*. *hirae*	−	−	−
KACC 16328	*E*. *hirae*	*E*. *hirae*	−	−	−
KACC 13883	*E*. *malodoratus*	*E*. *malodoratus*	−	−	−
KACC 13824	*E*. *mundtii*	*E*. *mundtii*	−	−	−
KCTC 3630	*E*. *mundtii*	*E*. *mundtii*	−	−	−
KACC 13781	*E*. *pseudoavium*	*E*. *pseudoavium*	−	−	−
KACC 13782	*E*. *raffinosus*	*E*. *raffinosus*	−	−	−
KACC 10789	*E*. *saccharolyticus*	*E*. *saccharolyticus*	−	−	−

**Table 3 foods-11-01046-t003:** Identification rates based on the BioTyper database with/without an in-house database.

Strains (No. of Strains)	No. of Strains with Results		
	≥2.300 log (Score)	2.000–2.299 log (Score)	1.700–1.999 log (Score)
Based on the BioTyper database			
Reference strains (5)	2 (40.0%)	3 (60.0%)	0 (0.0%)
Isolates (60)	4 (6.7%)	50 (83.3%)	6 (10.0%)
Total (65)	6 (9.2%)	53 (81.5%)	6 (9.2%)
Based on the in-house database			
Reference strains (5)	5 (100.0%)	0 (0.0%)	0 (0.0%)
Isolates (60)	54 (90.0%)	6 (10.0%)	0 (0.0%)
Total (65)	59 (90.8%)	6 (9.2%)	0 (0.0%)

**Table 4 foods-11-01046-t004:** Identification of isolates by BioTyper, in-house database, and specific mass peaks.

Strains	MALDI-TOF MS Database		Specific Peak (*m*/*z*)
	BioTyper (Score)	In-House Database (Score)	
SP1	*E*. *faecium* (2.157)	*E*. *lactis* (2.532)	*E*. *lactis* (3651.2, 7307.1)
SP2	*E*. *faecium* (2.205)	*E*. *lactis* (2.476)	*E*. *lactis* (3651.2, 7307.7)
SP3	*E*. *faecium* (2.2)	*E*. *lactis* (2.462)	*E*. *lactis* (3651.2, 7307.0)
SP4	*E*. *faecium* (2.241)	*E*. *lactis* (2.522)	*E*. *lactis* (3651.2, 7307.0)
SP5	*E*. *faecium* (2.171)	*E*. *lactis* (2.518)	*E*. *lactis* (3651.4, 7307.1)
SP6	*E*. *faecium* (2.178)	*E*. *lactis* (2.586)	*E*. *lactis* (3651.1, 7307.1)
SP7	*E*. *faecium* (2.171)	*E*. *lactis* (2.505)	*E*. *lactis* (3651.2, 7307.5)
SP8	*E*. *faecium* (2.227)	*E*. *lactis* (2.544)	*E*. *lactis* (3650.8, 7306.7)
SP9	*E*. *faecium* (1.992)	*E*. *lactis* (2.244)	*E*. *lactis* (3651.8, 7307.1)
SP10	*E*. *faecium* (2.087)	*E*. *lactis* (2.409)	*E*. *lactis* (3651.4, 7307.7)
SP11	*E*. *faecium* (2.128)	*E*. *lactis* (2.427)	*E*. *lactis* (3651.7, 7307.1)
SP12	*E*. *faecium* (1.978)	*E*. *lactis* (2.332)	*E*. *lactis* (3651.2, 7307.8)
SP13	*E*. *faecium* (2.231)	*E*. *lactis* (2.52)	*E*. *lactis* (3651.2, 7307.5)
RM1	*E*. *faecium* (2.088)	*E*. *lactis* (2.469)	*E*. *lactis* (3651.3, 7307.0)
RM2	*E*. *faecium* (2.193)	*E*. *lactis* (2.424)	*E*. *lactis* (3651.0, 7307.6)
RM3	*E*. *faecium* (2.186)	*E*. *lactis* (2.436)	*E*. *lactis* (3651.2, 7307.4)
RM4	*E*. *faecium* (1.996)	*E*. *lactis* (2.494)	*E*. *lactis* (3651.5, 7306.7)
RM5	*E*. *faecium* (2.212)	*E*. *lactis* (2.495)	*E*. *lactis* (3651.2, 7305.0)
RM6	*E*. *faecium* (2.012)	*E*. *lactis* (2.52)	*E*. *lactis* (3651.1, 7307.6)
RM7	*E*. *faecium* (2.216)	*E*. *lactis* (2.524)	*E*. *lactis* (3651.1, 7307.4)
RM8	*E*. *faecium* (2.1)	*E*. *lactis* (2.496)	*E*. *lactis* (3650.9, 7306.6)
RM9	*E*. *faecium* (2.099)	*E*. *lactis* (2.47)	*E*. *lactis* (3651.6, 7305.0)
RM10	*E*. *faecium* (2.142)	*E*. *lactis* (2.519)	*E*. *lactis* (3651.0, 7307.7)
RM11	*E*. *faecium* (2.15)	*E*. *lactis* (2.522)	*E*. *lactis* (3650.7, 7306.8)
RM12	*E*. *faecium* (2.061)	*E*. *lactis* (2.515)	*E*. *lactis* (3651.2, 7307.6)
RM13	*E*. *faecium* (2.122)	*E*. *lactis* (2.538)	*E*. *lactis* (3651.2, 7307.6)
RM14	*E*. *faecium* (2.103)	*E*. *lactis* (2.475)	*E*. *lactis* (3650.9, 7307.4)
RM15	*E*. *faecium* (2.193)	*E*. *lactis* (2.59)	*E*. *lactis* (3651.1, 7307.8)
PP1	*E*. *faecium* (2.226)	*E*. *faecium* (2.272)	*E*. *faecium* (10,124.6)
PP2	*E*. *faecium* (2.288)	*E*. *lactis* (2.28)/*E*. *faecium* (2.275)	*E*. *faecium* (10,124.3)
PP3	*E*. *faecium* (2.249)	*E*. *lactis* (2.209)/*E*. *faecium* (2.163)	*E*. *faecium* (10,123.6)
SS1	*E*. *faecium* (2.232)	*E*. *faecium* (2.342)	*E*. *faecium* (10,124.2)
SS2	*E*. *faecium* (2.351)	*E*. *lactis* (2.387)/*E*. *faecium* (2.37)	*E*. *faecium* (10,120.6)
SS3	*E*. *faecium* (2.21)	*E*. *lactis* (2.227)/*E*. *faecium* (2.214)	*E*. *faecium* (10,123.2)
SS4	*E*. *faecium* (2.279)	*E*. *lactis* (2.413)/*E*. *faecium* (2.259)	*E*. *faecium* (10,120.1)
SS5	*E*. *faecium* (2.381)	*E*. *faecium* (2.348)	*E*. *faecium* (10,120.0)
SS6	*E*. *faecium* (2.405)	*E*. *lactis* (2.332)/*E*. *faecium* (2.319)	*E*. *faecium* (10,121.3)
SS7	*E*. *faecium* (2.381)	*E*. *lactis* (2.205)/*E*. *faecium* (2.16)	*E*. *faecium* (10,124.3)
SS8	*E*. *faecium* (2.156)	*E*. *lactis* (2.435)	*E*. *lactis* (3650.7, 7306.3)
SS9	*E*. *faecium* (2.166)	*E*. *lactis* (2.444)	*E*. *lactis* (3650.3, 7305.8)
SS10	*E*. *faecium* (1.946)	*E*. *lactis* (2.336)	*E*. *lactis* (3651.0, 7307.5)
SS11	*E*. *faecium* (2.092)	*E*. *lactis* (2.522)	*E*. *lactis* (3650.8, 7307.0)
SS12	*E*. *faecium* (2.147)	*E*. *lactis* (2.378)	*E*. *lactis* (3650.9, 7306.5)
SS13	*E*. *faecium* (2.037)	*E*. *lactis* (2.514)	*E*. *lactis* (3651.1, 7307.7)
SS14	*E*. *faecium* (2.192)	*E*. *lactis* (2.371)	*E*. *lactis* (3650.5, 7306.1)
SS15	*E*. *faecium* (2.165)	*E*. *lactis* (2.548)	*E*. *lactis* (3649.7, 7305.9)
SS16	*E*. *faecium* (2.115)	*E*. *lactis* (2.501)	*E*. *lactis* (3651.2, 7305.9)
SS17	*E*. *faecium* (2.158)	*E*. *lactis* (2.436)	*E*. *lactis* (3649.8, 7306.6)
SS18	*E*. *faecium* (2.097)	*E*. *lactis* (2.44)	*E*. *lactis* (3650.2, 7305.2)
SS19	*E*. *faecium* (1.927)	*E*. *lactis* (2.337)	*E*. *lactis* (3650.7, 7307.6)
SS20	*E*. *faecium* (2.085)	*E*. *lactis* (2.39)	*E*. *lactis* (3650.2, 7306.0)
SS21	*E*. *faecium* (2.225)	*E*. *lactis* (2.592)	*E*. *lactis* (3650.1, 7305.3)
GS1	*E*. *faecium* (2.149)	*E*. *lactis* (2.482)	*E*. *lactis* (3650.1, 7305.1)
GS2	*E*. *faecium* (2.069)	*E*. *lactis* (2.445)	*E*. *lactis* (3649.9, 7305.3)
GS3	*E*. *faecium* (2.023)	*E*. *lactis* (2.53)	*E*. *lactis* (3649.8, 7305.6)
GS4	*E*. *faecium* (1.975)	*E*. *lactis* (2.503)	*E*. *lactis* (3650.2, 7306.6)
GS5	*E*. *faecium* (2.194)	*E*. *lactis* (2.463)	*E*. *lactis* (3650.0, 7305.2)
GS6	*E*. *faecium* (2.084)	*E*. *lactis* (2.32)	*E*. *lactis* (3650.2, 7305.6)
GS7	*E*. *faecium* (2.036)	*E*. *lactis* (2.537)	*E*. *lactis* (3650.1, 7306.0)
GS8	*E*. *faecium* (2.092)	*E*. *lactis* (2.504)	*E*. *lactis* (3650.2, 7306.3)

**Table 5 foods-11-01046-t005:** Identification of isolates using 16S rRNA gene sequencing.

Strains (No.)	Source	16S rRNA Gene Sequencing	
		Description	Accession No. (% Identity)
SP1-SP13 (13)	Soybean paste	*E*. *faecium*/*E*. *lactis*	MT597585.1/MG948154.1 (99.8%)
RM1-RM-15 (15)	Raw milk	*E*. *faecium*/*E*. *lactis*	MT597585.1/MG948154.1 (99.9%)
PP1-PP3 (3)	Probiotic	*E*. *faecium*	FJ378693.1 (99.0%)
SS1-SS3 (3)	Soy sauce	*E*. *faecium*	MN401132.1 (99.9%)
SS4-SS7 (4)	Soy sauce	*E*. *faecium*	MH473158.1 (99.9%)
SS8-SS15 (8)	Soy sauce	*E*. *faecium*/*E*. *lactis*	MT597585.1/MG948154.1 (100%)
SS16-SS21 (6)	Soy sauce	*E*. *faecium*/*E*. *lactis*	MT378127.1/CP082267.1 (99.9%)
GS1-GS8 (8)	Sikhae	*E*. *faecium*/*E*. *lactis*	MT597585.1/MG948154.1 (100%)

## Data Availability

The data presented in this study are available on request from the corresponding author.

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
