# Peer review of "Differentiating between *Enterococcus"

_foods, 2022, doi:10.3390/foods11071046_

Round 1

Reviewer 1 Report

Dear Authors,

The aim of this study was to evaluate the ability of MALDI-TOF mass spectrometry to distinguish two closely related Enterococcus species, for provided a viable alternative to 16S rRNA sequencing.

MALDI-TOF mass spectrometry is currently an increasingly important technique, especially in the clinical setting, to speed up bacterial identification and therefore anticipate clinical management and treatment of the patient. Therefore, studies related to this technique constitute an added value to the scientific literature.

I think this work is interesting and well written.

I kindly ask you to consider my comments/suggestions to improve your manuscript.

Best regards.

  • Title: Replace “Differentiating between Enterococcus faecium and Enterococcus lactis using mass peak analysis via matrix-assisted laser desorption ionization time-of-flight mass spectrometry” with “Differentiating between Enterococcus faecium and Enterococcus lactis by matrix-assisted laser desorption ionization time-of-flight mass spectrometry”.

  • The introduction is well written. However, at the lines 68-69, you could add that the BRUKER database currently lacks lactis reference spectra.

  • Line 34: Replace "however" with "but".

  • Line 47: After “time-consuming”, add "..,expensive,…".

  • Line 72: Replace “including” with "of which”.

  • Lines 72-75: Add the number of strains of each species analyzed. For example: n = 2 Enterococcus avium, etc.

  • Line 81 - Table 1: In tab.1, remove the superscript from the first acronyms or add it to all.

  • Move the table 1 after paragraph 2.1.

  • Why did you choose to analyze the isolates after 48h and not after 24h? Have you tried to evaluate if there was a variation in the spectrum between that obtained after 24h and that after 48h?

  • Line 97: Replace “to completely remove the ethanol” with “and supernatant was removed”.

  • Line 101-102: Remove “in 50% acetonitrile and 2.5% trifluoroacetic acid”.

  • Write "Bruker Daltonics, Bremen, Germany" when you mention it (lines 101, 104, 105, etc.).

  • Line 106: Replace “collected” with “obtained”.

  • Line 135: Remove “(5,628 reference spectra)”.

  • Line 136: Clarify that the MALDI-TOF MS analysis results are generally expressed with a score value, indicative of the matching between the sample spectrum and the reference spectra in database.

  • In the materials and methods, insert a paragraph that also describes 16S rRNA sequencing as it was the technique used to compare the results.

  • Figure 1: Since in Table 2 you describe in detail the presence / absence of three mass peaks in all the reference strains analyzed, in Fig. 1 you can add a mass spectrum of each analyzed species.

  • Figure 2: Was any other software used to generate Fig. 2? If so, please specify it.

  • Table 3: Add "log (score)" after the values of the second row.

  • Line 267: Replace “may be used” with “could be used”.

Author Response

Title: Replace “Differentiating between Enterococcus faecium and Enterococcus lactis using mass peak analysis via matrix-assisted laser desorption ionization time-of-flight mass spectrometry” with “Differentiating between Enterococcus faecium and Enterococcus lactis by matrix-assisted laser desorption ionization time-of-flight mass spectrometry”.

Response: Thank you for your critical comments. As you recommended, we replaced the Title.

Lines 2-4: Differentiating between Enterococcus faecium and Enterococcus lactis by matrix-assisted laser desorption ionization time-of-flight mass spectrometry

The introduction is well written. However, at the lines 68-69, you could add that the BRUKER database currently lacks lactis reference spectra.

Response: As you recommended, we added the sentence in line 69 as follows:

Line 69: BioTyper database currently lacks E. lactis reference spectra.

Line 34: Replace "however" with "but".

Response: As you recommended, we replaced “however” with “but” in line 34 as follows:

Line 34: have great potential as probiotics; but,

Line 47: After “time-consuming”, add "..,expensive,…".

Response: As you recommended, we added “expensive” in line 47 as follows:

Line 47: time-consuming, expensive, and requires additional analysis steps

Line 72: Replace “including” with "of which”.

Response: As you recommended, we replaced “including” with “of which” in line 74 as follows:

Line 74: Enterococcus reference strains, of which

Lines 72-75: Add the number of strains of each species analyzed. For example: n = 2 Enterococcus avium, etc.

Response: As you recommended, we added the number of strains of each species analyzed in lines 74-78 as follows:

Lines 74-78: E. avium (n = 2), E. casseliflavus (n = 1), E. devriesei (n = 1), E. durans (n = 1), E. faecalis (n = 3), E. faecium (n = 2), E. gallinarum (n = 1), E. gilvus (n = 1), E. hirae (n = 4), E. lactis (n = 3), E. malodoratus (n = 1), E. mundtii (n = 2), E. pseudoavium (n = 1), E. raffinosus (n = 1), and E. saccharolyticus (n = 1), were used (Table 1).

Line 81 - Table 1: In tab.1, remove the superscript from the first acronyms or add it to all.

Response: As you recommended, we removed the superscript from the first acronyms in Table 1.

Move the table 1 after paragraph 2.1.

Response: As you recommended, we moved Table 1 after paragraph 2.1.

Why did you choose to analyze the isolates after 48h and not after 24h? Have you tried to evaluate if there was a variation in the spectrum between that obtained after 24h and that after 48h?

Response: In the previous studies, in order to identify Enterococcus species by MALDI-TOF MS, isolates were cultured at 37°C for 48 h (Stępień-Pyśniak et al., 2017; Florio et al., 2019). A previous study has reported that the spectrum obtained after 48 h for anaerobes has a higher quality than that of 24 h (Veloo et al., 2014). Therefore, we analyzed isolates after 48 h. As you recommended, we added the references in line 83 as follows:

Line 83: Reference strains were grown under anaerobic conditions on de Man, Rogosa, and Sharpe agar (MRS, Difco, Becton & Dickinson, Sparks, MD, USA) for 48 h at 37°C [19–21].

Line 97: Replace “to completely remove the ethanol” with “and supernatant was removed”.

Response: As you recommended, we replaced “to completely remove the ethanol” with “and supernatant was removed” in line 110 as follows:

Line 110: for 10 min and supernatant was removed.

Line 101-102: Remove “in 50% acetonitrile and 2.5% trifluoroacetic acid”.

Response: As you recommended, we removed “in 50% acetonitrile and 2.5% trifluoroacetic acid”

Write "Bruker Daltonics, Bremen, Germany" when you mention it (lines 101, 104, 105, etc.).

Response: As you recommended, we added “Bruker Daltonics, Bremen, Germany” in lines 114, 117, 123, 126, 130, 136, and 147 as follows:

Lines 114, 117, 123, 126, 130, 136, and 147: Bruker Daltonics, Bremen, Germany

Line 106: Replace “collected” with “obtained”.

Response: As you recommended, we replaced “collected” with “obtained” in line 118 as follows:

Line 118: Data were obtained

Line 135: Remove “(5,628 reference spectra)”.

Response: As you recommended, we removed “(5,628 reference spectra)”.

Line 136: Clarify that the MALDI-TOF MS analysis results are generally expressed with a score value, indicative of the matching between the sample spectrum and the reference spectra in database.

Response: As you recommended, we added the sentence in lines 150-152 as follows:

Lines 150-152: The MALDI-TOF MS analysis results are generally expressed with a score value, indicative of the matching between the sample spectrum and the reference spectra in database.

In the materials and methods, insert a paragraph that also describes 16S rRNA sequencing as it was the technique used to compare the results.

Response: As you recommended, we added a paragraph for 16S rRNA sequencing in lines 94-104 as follows:

Lines 94-104: 2.2. 16S rRNA sequencing
The 16S rRNA sequencing of isolates was performed to compare the MALDI-TOF MS results. Genomic DNA of isolates was extracted using G-spin genomic DNA extraction kit (Intron Biotechnology, Seongnam, Korea). The amplification was carried out in the 25 µl mixture containing 2.5 mM dNTPs (Takara, Tokyo, Japan), 10× buffer (Takara), 0.5 units Ex Taq polymerase (Takara), 20 ng of template, and 400 nM of 27F/1492R primer set. The PCR thermal profile was performed at 95°C for 5 min, followed by 30 cycles of 95°C for 1 min, 58°C for 1 min, and 72°C for 2 min, and concluded with a final elongation at 72°C for 10 min. The PCR product was purified using the QIAquick PCR purification kit (Qiagen, Hilden, Germany) and sequenced. The 16S rRNA sequences of isolates were analyzed using the BLAST program.

Figure 1: Since in Table 2 you describe in detail the presence / absence of three mass peaks in all the reference strains analyzed, in Fig. 1 you can add a mass spectrum of each analyzed species.

Response: As you recommended, we newly added the mass spectrum of each analyzed species and added sentence in lines 168-169 and 287-290 as follows:

Lines 168-169: The mass spectra of each analyzed strain for non-target species are shown in Table S1.

Lines 287-290: Figure S1: The mass spectra of reference strains of E. avium, E. casseliflavus, E. devriesei, E. durans, E. faecalis, E. gallinarum, E. gilvus, E. hirae, E. malodoratus, E. mundtii, E. pseudoavium, E. raffinosus, E. saccharolyticus; m/z, mass-to-charge ratio; a.u., arbitrary units.

Figure 2: Was any other software used to generate Fig. 2? If so, please specify it.

Response: As you recommended, we added the sentence in lines 190-191 as follows:

Lines 190-191: Figure was generated using FlexAnalysis software version 3.4.

Table 3: Add "log (score)" after the values of the second row.

Response: As you recommended, we added “log (score)” after the values of the second row in Table 3.

Line 267: Replace “may be used” with “could be used”.

Response: As you recommended, we replaced “may be used” with “could be used” in line 286 as follows:

Line 286: could be used in quality control

Reviewer 2 Report

The current manuscript explores the concept of using MALDI-TOF-MS to differentiate two closely related Enterococcus species. Enterococcus faecium has been reported in contrast to Enterococcus lactis to be associated with human infection, virulence factors, and antibiotic resistance, whereas the later one is considered to be a potential probiotic candidate.  These two species are difficult to distinguish via conventional taxonomic methods and even the gold standard using 16S rRNA sequencing analyses shows limited discriminatory power for that purpose. MALDI-TOF MS is a powerful tool that utilizes protein based mass spectrometry data comparison to distinguish between bacterial species. The use of the technique demands a good sample preparation and an adequate database, which needs to be supplemented as in the present special case. Thereafter, the present study utilizes essentially a well prepared background serving as an appropriate basis for deriving the clearly defined objectives of the present study. The methods used are adequate, they represent standard sample preparation methods generally applied and are technically sound with certain parts (extension of the databank) improved. The results are presented compactly/concisely; the discussion corresponds to the data presented. Overall, the manuscript represents a straight forward study giving an interesting update to the “state of art” in this particular field, the authors have given altogether a good report confining to essential findings. The study would have gained more scientific input if the allocated specific mass peaks differentiating the two species could have been allocated and the proteins involved identified, thus serving as well defined markers.

Specific comments: 

Line 104: Analyses were performed via a Microflex LT bench-top mass spectrometer ….. Repetition!

Line 120: … in section 2.2 identifying….

Table 3: Please check the formatting, since some columns/cells have shifted and need to be aligned.

Line 246: The higher the throughput rate of a sample is, the lower is the 246 analysis cost/isolate…

Line 260: … this approach could be considered more efficient and accurate than the sometimes ambiguous, 16S rRNA sequencing.  Please do re-write using an alternative for “ambiguous” e.g. “lacking discriminating power of”

Line 264: The identification is based on mass spectrometric data and not on proteomic identification……

Author Response

Specific comments:

Line 104: Analyses were performed via a Microflex LT bench-top mass spectrometer ….. Repetition!

Response: As you recommended, we removed “Analyses were performed via a Microflex LT bench-top mass spectrometer”.

Line 120: … in section 2.2 identifying….

Response: As you recommended, we revised the sentence in line 134 as follows:

Line 134: in section 2.2 identifying

Table 3: Please check the formatting, since some columns/cells have shifted and need to be aligned.

Response: As you recommended, we shifted some columns and cells in Table 3.

Line 246: The higher the throughput rate of a sample is, the lower is the analysis cost/isolate…

Response: As you recommended, we revised the sentence in lines 264-265 as follows:

Lines 264-265: The higher the throughput rate of a sample is, the lower is the analysis cost/isolate

Line 260: … this approach could be considered more efficient and accurate than the sometimes ambiguous, 16S rRNA sequencing. Please do re-write using an alternative for “ambiguous” e.g. “lacking discriminating power of”

Response: As you recommended, we revised the sentence in lines 278-279 as follows:

Lines 278-279: this approach could be considered more efficient and accurate than the lacking discriminating power of 16S rRNA sequencing

Line 264: The identification is based on mass spectrometric data and not on proteomic identification……

Response: As you recommended, we revised the sentence in line 282 as follows:

Line 282: Moreover, the identification based on mass spectrometric data of two species